# Masked Random Noise for Communication-Efficient Federated Learning

Shiwei Li*
Huazhong University of Science and Technology
Wuhan, China
lishiwei@hust.edu.cn

Yingyi Cheng
Beijing University of Posts and Telecommunications
Beijing, China
leocheng@bupt.edu.cn

Haozhao Wang†
Huazhong University of Science and Technology
Wuhan, China
hz_wang@hust.edu.cn

Xing Tang
FiT, Tencent
Shenzhen, China
xing.tang@hotmail.com

Shijie Xu
Weihong Luo
FiT, Tencent
Shenzhen, China
shijiexu@tencent.com
lobbyluo@tencent.com

Yuhua Li
Huazhong University of Science and Technology
Wuhan, China
idcliyuhua@hust.edu.cn

Dugang Liu
Guangdong Laboratory of Artificial Intelligence and Digital Economy (SZ)
Shenzhen, China
dugang.ldg@gmail.com

Xiuqiang He
FiT, Tencent
Shenzhen, China
xiuqianghe@tencent.com

Ruixuan Li
Huazhong University of Science and Technology
Wuhan, China
rxli@hust.edu.cn

## ABSTRACT

Federated learning is a promising distributed training paradigm that effectively safeguards data privacy. However, it may involve significant communication costs, which hinders training efficiency. In this paper, we aim to enhance communication efficiency from a new perspective. Specifically, we request the distributed clients to find optimal model updates relative to global model parameters within predefined random noise. For this purpose, we propose **Federated Masked Random Noise (FedMRN)**, a novel framework that enables clients to learn a 1-bit mask for each model parameter and apply masked random noise (i.e., the Hadamard product of random noise and masks) to represent model updates. To make FedMRN feasible, we propose an advanced mask training strategy, called progressive stochastic masking (*PSM*). After local training, each client only need to transmit local masks and a random seed to the server. Additionally, we provide theoretical guarantees for the convergence of FedMRN under both strongly convex and non-convex assumptions. Extensive experiments are conducted on four popular datasets. The results show that FedMRN exhibits superior convergence speed and test accuracy compared to relevant baselines, while attaining a similar level of accuracy as FedAvg.

*This work was done when Shiwei Li worked as an intern at FiT, Tencent.
†Haozhao Wang is the corresponding author.

## CCS CONCEPTS

• **Computing methodologies → Distributed algorithms**.

## KEYWORDS

Federated Learning, Communication Efficiency, Supermasks

**ACM Reference Format:**
Shiwei Li, Yingyi Cheng, Haozhao Wang, Xing Tang, Shijie Xu, Weihong Luo, Yuhua Li, Dugang Liu, Xiuqiang He, and Ruixuan Li. 2024. Masked Random Noise for Communication-Efficient Federated Learning. In *Proceedings of the 32nd ACM International Conference on Multimedia (MM '24), October 28-November 1, 2024, Melbourne, VIC, Australia.* ACM, New York, NY, USA, 9 pages. https://doi.org/10.1145/3664647.3680608

## 1 INTRODUCTION

Federated learning (FL) [24] is a distributed training framework designed to protect data privacy. It allows distributed clients to collaboratively train a global model while retaining their data locally. FL typically consists of four steps: (1) clients download the global model from a central server, (2) clients train the global model using their respective data, (3) clients upload the model updates (changes in model parameters) back to the server, and (4) the server aggregates these updates to generate a new global model. This cycle is repeated for several rounds until the global model converges.

However, the iterative transmission of model parameters introduces significant communication overhead, which may affect training efficiency. To reduce communication costs, existing methods focus on two aspects: model compression and gradient compression. The former directly compresses model parameters [11], while the latter is to apply compression techniques on model updates after local training, such as quantization [20, 30] and sparsification [1, 28]. Notably, model compression reduces the model size, thereby constraining its capacity and learning capabilities. Although gradient

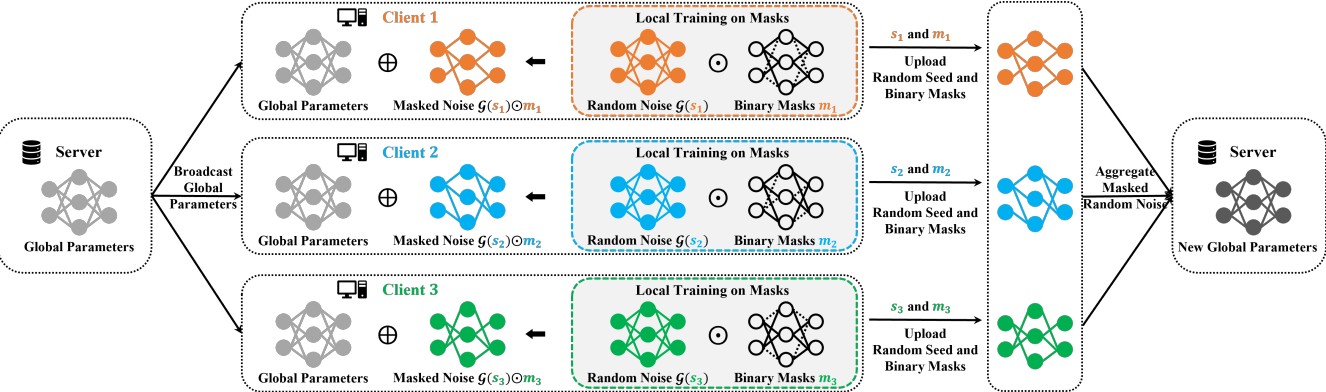

Figure 1: An illustration of FedMRN. $\mathcal{G}$ is a random noise generator. In the example, each binary mask $m \in \{0, 1\}$, therefore the masked random noise is sparse. It is worth noting that the mask can also take values from $\{-1, 1\}$, i.e., the signed mask. In such case, the presence of a dotted line indicates changing the sign of the corresponding noise, rather than pruning it off.

compression does not affect the model size, it can introduce notable errors into model updates, thereby influencing the convergence.

In this paper, we propose Federated Masked Random Noise (**FedMRN**) to compress the uplink communication where clients send model updates to the server. As shown in Figure 1, FedMRN requests clients to find optimal model updates within predefined random noise. In other words, model updates are now represented by the masked random noise, which is the Hadamard product of the random noise and binary masks. Note that the noise is determined by a specific random seed. The only trainable variables are the binary masks, which occupy 1 bit per parameter (bpp). Our **motivation** for doing this is twofold. First, recent studies [2, 29, 42] have confirmed the existence of the supermasks. These masks, when multiplied with randomly initialized weights, yield a model capable of achieving satisfactory accuracy without ever training the weights. To be more specific, finding masks for random noise can achieve comparable performance to training the parameters directly. Second, when it comes to FL, the objective of local training is to learn the optimal model updates relative to the global model parameters provided by the server. Therefore, we propose applying the concept of supermasks to the learning of model updates. We believe that training masks for random noise and generating masked random noise to serve as model updates can be as effective as directly training model parameters and then generating model updates. In doing so, we can compress the uplink communication overhead by a factor of 32, while neither reducing the number of model parameters nor requiring post-training compression on model updates.

To find the optimal masks, we maintain an extra learnable copy of model parameters within the local model, initialized with zeros to represent model updates. The masks will then be generated on the fly according to the random noise and model updates. In this way, we can optimize the masks indirectly by optimizing the above model updates. The generated masks will be used to map model updates into masked random noise, this process is called masking. To improve the accuracy of FedMRN, we further propose an advanced masking strategy, called progressive stochastic masking (*PSM*). *PSM* comprises two components: stochastic masking (*SM*) and progressive masking (*PM*). *SM* determines the probability of a mask being

set to 1 based on the model update and the corresponding random noise. It subsequently gets the mask by Bernoulli sampling with such probability. Further, *PM* probabilistically determines whether to perform masking on a model update. Specifically, during local training, each element within the model updates has a probability of being mapped into masked random noise. This probability gradually increases to 1 during local training, ensuring that all model updates will eventually be mapped into masked random noise.

The main contributions of this paper are summarized as follows:

- We propose a novel framework for communication-efficient FL, termed FedMRN, in which clients are requested to find optimal model updates within predefined random noise. In the uplink stage, FedMRN enables clients to transmit just a single 1-bit mask for each parameter to the server.
- We propose a local mask training strategy, dubbed progressive stochastic masking, aimed at finding the optimal masked random noise. The strategy consists of two key components: stochastic masking and progressive masking.
- We provide theoretical convergence guarantees for FedMRN under both strongly convex and non-convex assumptions, showing a comparable convergence rate to FedAvg [24].
- We perform extensive experiments on four datasets to compare FedMRN with relevant baselines. Experimental results demonstrate that FedMRN exhibits superior convergence speed and test accuracy compared to all baselines, while attaining a similar level of accuracy as FedAvg. The code is available at https://github.com/Leopold1423/fedmrn-mm24.

## 2 RELATED WORK

### 2.1 Lottery Tickets and Supermasks

The lottery ticket hypothesis (LTH) [6] states that within a neural network lie sparse subnetworks (aka winning tickets), capable of achieving comparable accuracy to the fully trained dense network when trained from scratch. Follow up work by Zhou et al. [42] finds that winning tickets perform far better than chance even without training. Inspired by this observation, Zhou et al. [42] propose identifying the supermasks, i.e., masks that can be applied to an

untrained network to produce a model with impressive accuracy. Specifically, they learn a probability $p$ for the mask of each randomly initialized and frozen weight. In the forward pass, each mask is obtained by Bernoulli sampling according to the corresponding probability $p = sigmoid(s)$, where $s$ is the trainable variable and will be optimized by gradient descent. Subsequently, Koster et al. [16] have extended the value range of the mask to $\{-1, 0, 1\}$. These three values represent sign-inverting, dropping, and keeping the weight, respectively. In summary, above studies as well recent theoretical results [23, 27] indicate that masking a randomly initialized network can be as effective as directly training its weights.

## 2.2 Supermasks for Federated Learning

Recent studies [13, 18, 35] have employed supermasks to reduce communication costs for FL. They attempt to find supermasks for randomly initialized network in federated settings. Note that the mask $m$ is typically optimized by training a continuous score $s$. Specifically, FedMask [18] gets the mask by $m = \mathbb{I}(sigmoid(s) > 0.5)$, utilizing the indicator function $\mathbb{I}$. FedPM [13] generates the mask through Bernoulli sampling, that is, $m = Bern(sigmoid(s))$. HideNseek [35] changes the value range of the mask to $\{-1, 1\}$, where the mask is generated by $m = 2\mathbb{I}(s > 0) - 1$.

Logically speaking, in FL, we should directly communicate and aggregate the trainable parameters, i.e., the scores $s$. However, in order to reduce communication costs, the above methods request clients to upload only the binary masks generated from the scores $s$. The server will aggregate the masks and then generate an estimate for $s$. For example, $s^{t+1} = sigmoid^{-1}[\frac{1}{K}\sum_{k=1}^{N} Bern(sigmoid(s_k^t))]$ in FedPM. Essentially, it is a form of model compression, where the trainable scores are binarized into masks through Bernoulli sampling. This will introduce significant errors to the updating of the scores and hamper their stable and effective training.

We analyze that the above methods are very rough combinations of the supermasks and FL. They are stuck in the inertia of finding random subnetworks. FL encompasses two levels: (1) the entire training process, whose goal is to find the optimal global parameters, and (2) the local training process, whose goal is to find the optimal model updates relative to the currently received global parameters. To integrate supermasks into the former level, we should leverage the entire training process to train the scores of masks. In this case, model updates are the changes of scores. Consequently, compressing these scores into masks for communication introduces significant errors, thereby affecting their optimization. In this paper, we propose to combine the supermasks with the latter level, i.e., local training, and try to learn model updates with supermasks. Specifically, we train masks for random noise and generate masked random noise to serve as model updates. In doing so, model updates are no longer changes in scores, but directly the values of masks. In summary, the biggest difference between our work and existing studies is that we find masked random noise to serve as model updates of local training, rather than as the final parameters.

## 2.3 Communication Compression

The first way of communication compression is model compression. For example, FedPara [11] performs low-rank decomposition on weight matrices. FedSparsify [33] performs weight magnitude pruning during local training and only parameters with larger magnitude will be sent to the server. These methods often have other advantages besides efficient communication, such as smaller model storage and less computational overhead. Nevertheless, their capacity to compress communication costs is frequently restricted. Intense compression markedly diminishes the size of local models, consequently undermining their learning ability.

Another way of communication compression is gradient compression, such as sparsification [1, 28] and quantization [10, 14, 20, 30]. ZeroFL [28] prunes model updates based on the magnitude with a given sparsity ratio. FedPAQ [30] propose to quantize local model updates. [3, 14, 15, 34] has further investigated the application of 1-bit quantization (i.e., binarization) on model updates. Moreover, DRIVE [37] and EDEN [36] use shared randomness to improve the model accuracy of binarization and quantization. Let $x \in \mathbb{R}^d$ denote the vector to be compressed and $R$ be a matrix generated with a random seed. DRIVE compressed $x$ into $\hat{x} = \alpha R^{-1}sign(Rx)$. Each client only needs to upload the scalar $\alpha$ and the signs of $Rx$ to the server. Note that the scalar $\alpha$ is calculated by clients to minimize $\|x - \hat{x}\|$. Later, EDEN has extended DRIVE to the case of quantization and further improves the calculation process for the scalar $\alpha$. However, existing gradient compression methods are compressing model updates in a post-training manner. In contrast, we directly associate model updates with masked random noise during local training, which is a form of learning to compress model updates. This allows us to minimize the impact of errors caused by model updates compression on accuracy through local training.

# 3 METHODOLOGY

## 3.1 Problem Formulation

FL involves $N$ clients connecting to a central server. The general goal of FL is to train a global model by multiple rounds of local training on each client's local dataset, which can be formulated as:

$$\min_{\mathbf{w} \in \mathbb{R}^d} F(\mathbf{w}) = \sum_{k=1}^{N} p_k F_k(\mathbf{w}), \quad (1)$$

where $p_k$ is the proportion of the $k$-th client's data to all the data of the $N$ clients and $F_k$ is the objective function of the $k$-th client. FedAvg [24] is a widely used FL algorithm. In the $t$-th round, the server sends the global parameters $\mathbf{w}^t$ to several randomly selected $K$ clients. The set of selected clients can be denoted as $C_t$. Each selected client seeks to determine the optimal model updates relative to $\mathbf{w}^t$ through multiple gradient descents on its local dataset:

$$\min_{\mathbf{u}_k^t \in \mathbb{R}^d} F_k(\mathbf{w}^t + \mathbf{u}_k^t). \quad (2)$$

Note that $\mathbf{u}_k^t$ denotes the accumulation of updates produced by multiple gradient descents. After local training, each client obtains its local model updates $\mathbf{u}_k^t$ and then send them to the server, who then aggregates all the model updates to generate new global model parameters as follows:

$$\mathbf{w}^{t+1} = \mathbf{w}^t + \sum_{k \in C_t} p_k' \mathbf{u}_k^t. \quad (3)$$

where $p_k' = \frac{p_k}{\sum_{k \in C_t} p_k}$ denotes the proportion of the $k$-th client's data to all the data used in the $t$-th round.

In this paper, we suggest identifying the optimal masked random noise to serve as model updates relative to global parameters. Thus, the objective of local training can be formulated as follows:

$$\min_{\boldsymbol{m}_k^t \in \{0,1\}^d} F_k(\mathbf{w}^t + \mathcal{G}(s_k^t) \odot \boldsymbol{m}_k^t), \tag{4}$$

where $\odot$ is the Hadamard product, $\mathcal{G}$ is a random noise generator, $s_k^t$ is the random seed, and $\boldsymbol{m}_k^t$ denotes the masks. $\mathcal{G}$ can correspond to various data distributions, such as Gaussian, Uniform, and Bernoulli distributions. The optimization space of the mask can also be $\{-1, 1\}$, i.e., the signed mask. In fact, binary masks and signed masks are equivalent to a certain extent, as illustrated by the subsequent expansion: $\mathcal{G}(s) \odot \boldsymbol{m}_s = 2\mathcal{G}(s) \odot \boldsymbol{m} - \mathcal{G}(s)$, where $\boldsymbol{m}_s \in \{-1, 1\}^d$ and $\boldsymbol{m} \in \{0, 1\}^d$. It is evident that the noise required for the binary masks $\boldsymbol{m}$ is twice that of the signed masks $\boldsymbol{m}_s$.

After local training, each client only sends the random seed $s_k^t$ and the masks $\boldsymbol{m}_k^t$ to the server. Then the server will recover the masked random noise and performs central model aggregation by

$$\mathbf{w}^{t+1} = \mathbf{w}^t + \sum_{k \in C_t} p_k' \mathcal{G}(s_k^t) \odot \boldsymbol{m}_k^t. \tag{5}$$

In the following, we will elaborate on the optimization of local masks in Section 3.2, and introduce the pipeline of our federated training framework in Section 3.3.

## 3.2 Progressive Stochastic Masking

In the $t$-th round, the clients will receive global model parameters $\mathbf{w}^t$ from the server, and then employ them to initialize their local models. Next, each client generates its local noise by a random seed $s_k^t$, that is $\mathcal{G}(s_k^t)$. To find the optimal masks, we additionally maintain a trainable copy of model parameters within the local model, initialized with zeros to represent model updates. The masks will then be generated on the fly according to the random noise and model updates. In this way, the masks can be optimized indirectly by optimizing the model updates. The notation of model updates is $\boldsymbol{u}_k^{t,\tau}$, where $k$ represents the client serial number, $t$ denotes the current training round, and $\tau$ denotes the current local step.

The process of generating masks and obtaining masked random noise is termed masking. Next, we will introduce an effective mask training strategy for local training, called progressive stochastic masking (PSM). PSM includes the design of two components, namely stochastic masking (SM) and progressive masking (PM).

*3.2.1 Stochastic Masking.* A simple way to generate masks is by comparing the signs of model updates and random noise. Specifically, a mask should be set to 1 only when the model update and the corresponding noise share the same sign. We name this method deterministic masking (DM). There are many methods similar to DM, such as the sign function used in SignSGD [3] and the indicator function used in FedMask. However, DM suffers from a significant flaw, that is seriously biased estimation. For ease of expression, we next use $\hat{\boldsymbol{u}} = \mathcal{G}(s) \odot \boldsymbol{m}$ to represent the masked random noise. The biased estimation refers to the huge deviation between $\boldsymbol{u}$ and and the expectation of $\hat{\boldsymbol{u}}$. More realistically, DM disregards the amplitude of both model updates and the random noise.

To overcome this issue, we propose the stochastic masking strategy, as shown in Figure 2(a). Specifically, SM first calculates the probability that a mask is 1 based on the value of the model update

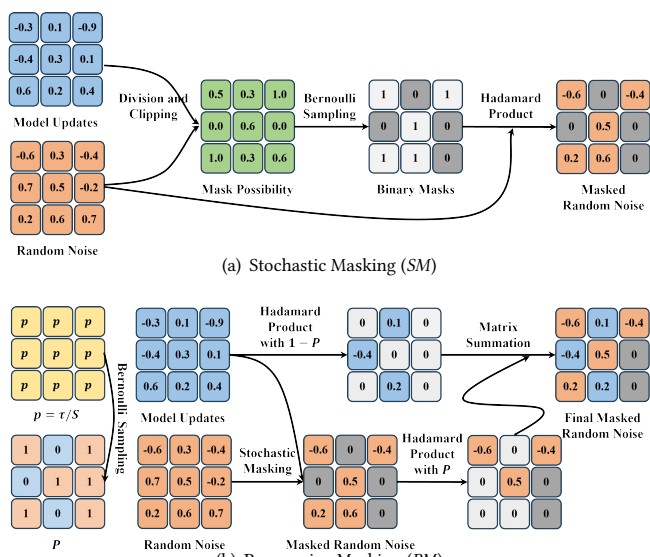

(a) Stochastic Masking (SM)

(b) Progressive Masking (PM)

**Figure 2: Schematic diagram of *SM* and *PM*. In subfigure (b), $\tau$ is the number of current local iterations, and $S$ is the total number of local iteration steps. $p$ will increase to 1 as training progresses, so that each element of the model updates will eventually be mapped into masked noise.**

and the random noise, and then uses this probability to sample the binary mask from the Bernoulli distribution. For a model update $u$ and the corresponding noise $n$, the probability and the binary mask has the following values:

$$m = \mathcal{M}(u, n) = \begin{cases} 1 & \text{w.p.} \quad p = clip(u/n, 0, 1), \\ 0 & \text{w.p.} \quad 1 - p, \end{cases} \tag{6}$$

where $clip$ is to clamp the probability into the range of $[0, 1]$. Obviously, $\mathbb{E}[n\mathcal{M}(u, n) - u] = 0$ when $u/n \in [0, 1]$. That is, the expectation of the error caused by masking is zero. Also, we extend our stochastic masking strategy to the signed masks, where the calculation of the probability and the mask shall be changed to

$$m = \mathcal{M}(u, n) = \begin{cases} 1 & \text{w.p.} \quad p = clip(u+n/2n, 0, 1), \\ -1 & \text{w.p.} \quad 1 - p. \end{cases} \tag{7}$$

In this case, $\mathbb{E}[n\mathcal{M}(u, n) - u] = 0$ when $u/n \in [-1, 1]$. Applying the mask generator $\mathcal{M}$ defined in Eq.(6) or (7) to element-wise process the model updates $\boldsymbol{u}$ and the random noise $\mathcal{G}(s)$, we can derive the formula for stochastic masking:

$$\hat{\boldsymbol{u}} = \mathcal{S}(\boldsymbol{u}, \mathcal{G}(s)) = \mathcal{G}(s) \odot \mathcal{M}(\boldsymbol{u}, \mathcal{G}(s)) \tag{8}$$

During forward propagation, the masked noise $\hat{\boldsymbol{u}}$ will be added to the frozen global parameters to obtain the complete model parameters. However, the stochastic masking $\mathcal{S}$ includes the process of Bernoulli sampling, which is non-differentiable. This renders the gradient descent algorithms inapplicable. To optimize $\boldsymbol{u}$ and thus optimizing the masks, we adopt Straight-through Estimator (STE) [8] to treat SM as an identity map during backpropagation, i.e.,

 

$\partial S / \partial u = 1$. Recent work has demonstrated that STE works as a first-order approximation of the gradient and affirmed its efficacy [22]. Hence, the model updates $u$ will be optimized by

$$u_k^{t,\tau+1} = u_k^{t,\tau} - \eta \frac{\partial F_k(\mathbf{w}^t + \hat{u}_k^{t,\tau})}{\partial \hat{u}_k^{t,\tau}}. \tag{9}$$

*3.2.2 Progressive Masking.* SM has addressed the issue of biased estimation caused by masking. Specifically, $\mathbb{E}[\hat{u} - u] = 0$ under certain conditions. However, there is still a certain gap between the model updates $u$ and the masked noise $\hat{u}$. As shown in Eq.(9), $u$ is essentially updated using the gradient of $\hat{u}$. This gap can significantly disrupt the optimization process for $u$, especially during the initial training stages when $u$ is far from stable convergence.

To reduce the gap between $u$ and $\hat{u}$ during optimization, we propose a progressive masking strategy. Figure 2(b) illustrates the process, where each model update undergoes a probability of being mapped into masked noise during forward propagation. As training proceeds, the probability will gradually increase to 1, ensuring that every element within the model updates will eventually be mapped into masked noise. Here, we simply define the probability to increase linearly, i.e., $p = \tau/S$, where $S$ is the total local iteration steps and $\tau$ is the serial number of the current step. The combination of *SM* and *PM* yields *PSM*, wherein the actual model updates utilized in the forward pass are

$$\hat{u} = (1 - P) \odot \bar{u} + P \odot S(u, \mathcal{G}(s)). \tag{10}$$

$P = Bern(1 \times p) \in \{0,1\}^d$, each element within $P$ is obtained by Bernoulli sampling with probability $p$. $\bar{u} = clip(u, \mathcal{G}(s))$, limiting $u$ to the interval of $[0, \mathcal{G}(s)]$, or $[\mathcal{G}(s), 0]$ where $\mathcal{G}(s)$ is negative. For the signed mask, the interval will be $[-|\mathcal{G}(s)|, |\mathcal{G}(s)|]$.

As both *SM* and *PM* employ Bernoulli sampling, we offer a comparison to alleviate potential confusion. Bernoulli sampling in *PM* determines whether to map a model update into masked noise, with the probability based on the local training progress. Differently, in *SM*, Bernoulli sampling determines whether to set a mask to 1, with the probability calculated by the values of model updates and noise.

## 3.3 Federated Masked Random Noise

In the preceding subsections, we described our objectives and elucidated the local training procedure. Hereafter, we present the comprehensive framework of FedMRN. The pipeline of FedMRN closely resembles that of FedAvg, with nuanced differences in the local training and central aggregation stages. Consequently, FedMRN can be seamlessly integrated into prevalent FL frameworks. The pipeline of FedMRN is detailed in Algorithm 1.

The server maintains a global model, whenever a new round of training starts, it sends the latest global parameters to randomly selected clients. These clients will load the global parameters, generate random noise, and initialize the local model updates with zeros. Subsequently, clients conduct local training using the PSM strategy. Line 15-18 in Algorithm 1 condenses all designs in Section 3.2. After local training, each client produces the final masks $\boldsymbol{m}_k^t$ and send them to the server along with the random seed $s_k^t$. Upon receiving these contents, the server will recover the model updates (i.e., the masked random noise) of each client and aggregate them using Eq.(5) to generate global model parameters for the next round.

---

**Algorithm 1 Federated Masked Random Noise**

**Input:** learning rate $\eta$; client data ratios $\{p_k | k \in [N]\}$; noise generator $\mathcal{G}$; mask generator $\mathcal{M}$.
**Output:** Trained global model $\mathbf{w}$.
1: Initialize the model parameters $\mathbf{w}^1$;
2: **procedure** SERVER-SIDE OPTIMIZATION
3:     **for** each communication round $t \in \{1, 2, ..., R\}$ **do**
4:         Randomly select a subset of clients $C_t$;
5:         Broadcast $\mathbf{w}^t$ to each selected client;
6:         **for** each selected client $k$ **in parallel do**
7:             $\boldsymbol{m}_k^t, s_k^t \leftarrow ClientLocalUpdate(\mathbf{w}^t)$;
8:         Aggregate masked random noise by
9:         $\mathbf{w}^{t+1} = \mathbf{w}^t + \frac{\sum_{k \in C_t} p_k \mathcal{G}(s_k^t) \odot \boldsymbol{m}_k^t}{\sum_{k \in C_t} p_k}$;
10: **procedure** CLIENTLOCALUPDATE($\mathbf{w}^t$)
11:     Load global model parameters $\mathbf{w}^t$;
12:     Generate noise by $\mathcal{G}$ with seed $s_k^t$;
13:     Initializes model updates $u_k^{t,1}$ with zeros;
14:     **for** each local iteration $\tau \in \{1, 2, ..., S\}$ **do**
15:         $\hat{u}_k^{t,\tau} = \mathcal{M}(u_k^{t,\tau}, \mathcal{G}(s_k^t)) \odot \mathcal{G}(s_k^t)$;    # SM
16:         $P = Bern(1 \times \tau/S)$;
17:         $\bar{u}_k^{t,\tau} = clip(u_k^{t,\tau}, \mathcal{G}(s_k^t))$;
18:         $\hat{u}_k^{t,\tau} = (1 - P) \odot \bar{u}_k^{t,\tau} + P \odot \hat{u}_k^{t,\tau}$;    # PM
19:         $u_k^{t,\tau+1} = u_k^{t,\tau} - \eta \frac{\partial F_k(\mathbf{w}^t + \hat{u}_k^{t,\tau})}{\partial \hat{u}_k^{t,\tau}}$
20:     **return** final masks $\mathcal{M}(u_k^{t,S+1}, \mathcal{G}(s_k^t))$ and the seed $s_k^t$.

---

## 4 CONVERGENCE ANALYSIS

In this section, we present our theoretical guarantees on the convergence of FedMRN, taking into account the non-independently identically distributed (Non-IID) nature of local datasets. For simplicity, we analyze the convergence of FedMRN using signed masks. We first consider the strongly convex setting and state the convergence guarantee of FedMRN for such losses in Theorem 1. Then, in Theorem 2, we present the overall complexity of FedMRN for finding a first-order stationary point of the global objective function $F$, when the loss function is non-convex. All proofs are provided in the Appendix.

Before that, we first give the following notations and assumptions required for convex and non-convex settings. Let $F^*$ and $F_k^*$ be the minimum values of $F$ and $F_k$, respectively. We use the term $\Gamma = F^* - \sum_{k=1}^N p_k F_k^*$ for quantifying the degree of data heterogeneity. In Section 3.3, the subscripts $t \in [R]$ and $\tau \in [S]$ are used to represent the serial number of global rounds and local iterations, respectively. In the following analysis, we will only use the subscript $t$ to represent the cumulative number of iteration steps in the sense that $t \in [T], T = RS$. Below are some commonly used assumptions:

ASSUMPTION 1. *(L-smoothness.)* $F_1, ..., F_N$ *are all L-smooth: for all* $\mathbf{w}$ *and* $\mathbf{v}$, $F_k(\mathbf{v}) \leq F_k(\mathbf{w}) + (\mathbf{v} - \mathbf{w})^T \nabla F_k(\mathbf{w}) + \frac{L}{2} \|\mathbf{v} - \mathbf{w}\|^2$.

ASSUMPTION 2. *(Bounded variance.) Let* $\xi_k^t$ *be sampled from the k-th client's local data randomly. The variance of stochastic gradients is bounded:* $\mathbb{E}\|\nabla F_k(\mathbf{w}_k^t, \xi_k^t) - \nabla F_k(\mathbf{w}_k^t)\| \leq \sigma$ *for all* $k = 1, ..., N$.

ASSUMPTION 3. *(Bounded gradient.) The expected squared norm of stochastic gradients is uniformly bounded, i.e.,* $\mathbb{E}\|\nabla F_k(\mathbf{w}_k^t, \xi_k^t)\| \leq G$ *for all* $k = 1, ..., N$ *and* $t = 1, ..., T$.

ASSUMPTION 4. *(Bounded error.) The error caused by the masking function* $\mathcal{S}$ *grows with the* $l_2$*-norm of its argument, i.e.,* $\mathbb{E}\|\mathcal{S}(x, \mathcal{G}(s)) - x\| \leq q\|x\|$.

ASSUMPTION 5. *(Strongly convex.)* $F_1, ..., F_N$ *are* $u$*-strongly convex: for all* $\mathbf{w}$ *and* $\mathbf{v}$, $F_k(\mathbf{v}) \geq F_k(\mathbf{w}) + (\mathbf{v} - \mathbf{w})^T \nabla F_k(\mathbf{w}) + \frac{\mu}{2}\|\mathbf{v} - \mathbf{w}\|^2$.

Assumptions 1-3 are commonplace in standard optimization analyses [21, 32]. The condition in Assumption 4 is satisfied with many compression schemes including the masking function $\mathcal{S}$ in Eq.(8). Assumption 4 is also used in [15, 30] to analyze the convergence of federated algorithms. Assumption 5 is about strong convexity and will not be used in the non-convex settings.

THEOREM 1. *(Strongly convex.) Let Assumptions 1-5 hold. Choose* $\kappa = L/\mu, \gamma = \max\{8\kappa, S\} - 1$ *and the learning rate* $\eta_t = 2/\mu(\gamma+t)$. *Generating the noise from the Bernoulli distribution* $\{-2\eta_0 SG, 2\eta_0 SG\}$, *then FedMRN satisfies*

$$\mathbb{E}[F(\mathbf{w}_T)] - F^* \leq \frac{\kappa}{\gamma + T}(\frac{2B}{\mu} + \frac{\mu(\gamma+1)}{2}\mathbb{E}\|\mathbf{w}_1 - \mathbf{w}^*\|^2), \quad (11)$$

*where* $B = \frac{\sigma^2}{N} + 6L\Gamma + 8(1+q^2)(S-1)^2 G^2 + 4\frac{q^2(N-1)+N-K}{K(N-1)}S^2 G^2$.

THEOREM 2. *(Non-convex.) Let Assumptions 1-4 hold. Assume the learning rate is set to* $\eta = \frac{1}{L\sqrt{T}}$. *Generating the noise from the Bernoulli distribution* $\{-2\eta SG, 2\eta SG\}$, *then the following first-order stationary condition holds*

$$\frac{1}{T}\sum_{t=0}^{T-1}\mathbb{E}\|\nabla F(\mathbf{w}_t)\|^2 \leq \frac{2L(F(\mathbf{w}_0) - F^*)}{\sqrt{T}} + \frac{P}{\sqrt{T}} + \frac{Q}{T}, \quad (12)$$

*where* $P = \frac{\sigma^2}{N} + 4\frac{q^2(N-1)+N-K}{K(N-1)}S^2 G^2)$ *and* $Q = 4(1+q^2)(S-1)^2 G^2$.

PROPOSITION 1. *In Theorems 1 and 2, for simplicity, we only consider the effect of SM and temporarily ignore PM. In fact, PM can further reduce* $q$ *by* $\sqrt{\frac{1}{S^3}\sum_{\tau=1}^{S}\tau^2}$ *times.*

REMARK 1. *By setting* $q = 0$, *Theorem 1 is equivalent to the analysis about FedAvg in [21]. By setting* $K = N$ *and* $S = 1$, *Theorems 1 and 2 recovers the convergence rate of SignSGD [31] when used in distributed training. By setting* $K = N, S = 1$ *and* $q = 0$, *Theorems 1 and 2 can recover the convergence rate of vanilla SGD.*

REMARK 2. *Under the conditions of Theorem 1 and 2, the convergence rate of both FedMRN and FedAvg (*$q = 0$*) is* $O(\frac{1}{T})$ *in the strongly convex setting, and* $O(\frac{1}{T}) + O(\frac{1}{\sqrt{T}})$ *in the non-convex setting.*

## 5 EXPERIMENTS

### 5.1 Experimental Setup

*5.1.1 Datasets and Models.* In this section, we evaluate FedMRN on four widely used datasets: FMNIST [39], SVHN [26], CIFAR-10 and CIFAR-100 [17]. For FMNIST and SVHN, we employ a convolutional neural network (CNN) with four convolution layers and one fully connected layer. For CIFAR-10 and CIFAR-100, we employ a CNN with eight convolution layers and one fully connected layer.

ReLU [7] is used as the activation function and batch normalization (BN) [12] is utilized to ensure stable training. Experiments on other tasks [5, 25] and models [9, 40] can be find in the Appendix.

*5.1.2 Data Partitioning.* We consider both cases of IID and Non-IID data distribution, referring to the data partitioning benchmark of FL [19]. Under IID partitioning, an equal quantity of data is randomly sampled for each client. The Non-IID scenario further encompasses two distinct label distributions, termed Non-IID-1 and Non-IID-2. In Non-IID-1, the proportion of the same label among clients follows the Dirichlet distribution [41], while in Non-IID-2, each client only contains data of partial labels. For CIFAR-100, we set the Dirichlet parameter to 0.2 in Non-IID-1 and assign 20 random labels to each client in Non-IID-2. For the other datasets, we set the Dirichlet parameter to 0.3 in Non-IID-1 and assign 3 random labels to each client in Non-IID-2.

*5.1.3 Baselines.* **FedAvg** [24] is adopted as the backbone training algorithm. We compare **FedMRN** with several state-of-the-art methods, including **FedPM** [13], **FedSparsify** [33], **SignSGD** [31], **Top-**$k$ [1], **TernGrad** [38], **DRIVE** [37], **EDEN** [36]. FedPM and FedSparsify focus on model compression, while the remaining baselines concentrate on gradient compression. FedPM trains and communicates a binary mask for each model parameter. FedSparsify prunes the model weights during local training with a specified sparsity ratio and finally uploads the pruned model. Similarly, Top-$k$ prunes model updates after local training by a sparsity ratio. SignSGD performs stochastic binarization on model updates, while TernGrad converts the model updates to ternary values. EDEN and DRIVE initially execute a random rotation on model updates (essentially multiplying by a random matrix) before binarizing them. The communication costs of FedPM, SignSGD, EDEN, DRIVE, and FedMRN are all 1 bit per parameter (bpp). Therefore, for a fair comparison, we set the sparsity of FedSparsify and Top-$k$ to 97%, resulting in approximately 32-fold compression. Note that we did not consider the extra overhead of sparse encoding. Otherwise, a higher sparsity would be required. Additionally, the communication costs of TernGrad is log(3) bpp, surpassing that of other methods.

*5.1.4 Hyperparameters.* The number of clients is set to 100 and 10 clients will be selected for training in each round. The local epoch is set to 10 and the batch size is set to 64. SGD [4] is used as the local optimizer. The learning rate is tuned from {1.0, 0.3, 0.1, 0.03, 0.01}. The number of rounds are set to 100 for FMNIST and SVHN, and are set to 200 for CIFAR-10 and CIFAR-100. Note that **FedMRN** and **FedMRNS** indicate the use of binary masks and signed masks, respectively. The random noise in FedMRN follows a uniform distribution by default. The range of the distribution is [-1e-2, 1e-2] for FedMRN and [-5e-3, 5e-3] for FedMRNS. Each experiment is run five times on Nvidia 3090 GPUs with Intel Xeon E5-2673 CPUs. Average results and the standard deviation are reported.

### 5.2 Overall Performance

In this subsection, we compare the performance of FedMRN and the baselines by the global model accuracy and the convergence speed. All numerical results are reported in Table 1. Further, to facilitate comparison of accuracy, the accuracy loss of each method relative to FedAvg is displayed in Table 2, where each term represents the

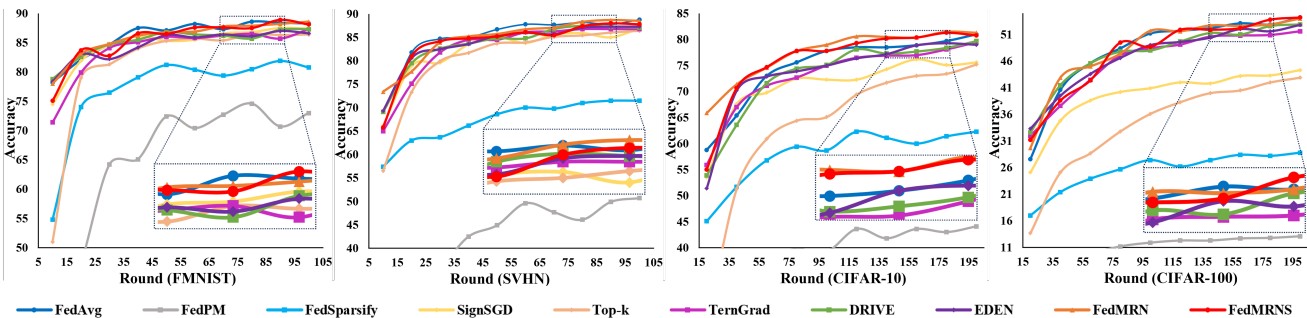

Figure 3: Convergence curves under the Non-IID-2 data distribution.

Table 1: The accuracy of all methods on four datasets. The best accuracy is bolded and the next best accuracy is underlined. FedMRN and FedMRNS indicate the use of binary masks $\{0, 1\}$ and signed masks $\{-1, 1\}$, respectively.

| | FMNIST | | | SVHN | | | CIFAR-10 | | | CIFAR-100 | | |
|---|---|---|---|---|---|---|---|---|---|---|---|---|
| | IID | Non-IID-1 | Non-IID-2 | IID | Non-IID-1 | Non-IID-2 | IID | Non-IID-1 | Non-IID-2 | IID | Non-IID-1 | Non-IID-2 |
| FedAvg | **92.0 (± 0.1)** | **90.5 (± 0.1)** | 88.8 (± 0.2) | 92.2 (± 0.1) | 90.3 (± 0.2) | **88.9 (± 0.1)** | 88.2 (± 0.2) | 84.1 (± 0.2) | 80.8 (± 0.5) | 56.1 (± 0.3) | 54.8 (± 0.5) | 54.4 (± 0.2) |
| FedPM | 81.7 (± 0.4) | 78.6 (± 0.2) | 74.6 (± 0.8) | 66.3 (± 0.9) | 52.4 (± 1.1) | 48.3 (± 2.3) | 50.2 (± 0.9) | 46.3 (± 0.2) | 44.3 (± 0.5) | 22.9 (± 0.1) | 16.6 (± 0.3) | 13.4 (± 0.2) |
| FedSparsify | 85.6 (± 0.1) | 80.9 (± 1.2) | 78.8 (± 1.5) | 81.3 (± 0.3) | 77.2 (± 0.4) | 72.8 (± 1.1) | 72 (± 0.7) | 66.8 (± 0.7) | 62.5 (± 0.5) | 32.9 (± 0.2) | 29.8 (± 0.3) | 28.8 (± 0.1) |
| SignSGD | 91.1 (± 0.1) | 88.4 (± 0.2) | 87.1 (± 0.3) | 90.8 (± 0.2) | 87.3 (± 0.4) | 86.5 (± 0.3) | 85.3 (± 0.2) | 75.1 (± 0.4) | 76.2 (± 0.6) | 48.0 (± 0.5) | 39.2 (± 0.2) | 44.0 (± 0.2) |
| Top-$k$ | 90.1 (± 0.1) | 88.6 (± 0.2) | 86.7 (± 0.2) | 90.0 (± 0.1) | 87.7 (± 0.1) | 86.4 (± 0.1) | 84.1 (± 0.1) | 77.9 (± 0.3) | 75.1 (± 0.1) | 50.2 (± 0.2) | 47.6 (± 0.5) | 43.0 (± 0.7) |
| TernGard | 91.4 (± 0.2) | 89.9 (± 0.2) | 87.9 (± 0.2) | 91.7 (± 0.1) | 89.6 (± 0.1) | 87.7 (± 0.3) | 86.9 (± 0.2) | 81.9 (± 0.5) | 79.2 (± 0.4) | 53.5 (± 0.4) | 52.1 (± 0.5) | 52.3 (± 0.3) |
| DRIVE | 91.6 (± 0.1) | 89.9 (± 0.2) | 88.1 (± 0.1) | 91.6 (± 0.2) | 89.6 (± 0.2) | 87.8 (± 0.3) | 87.2 (± 0.1) | 82.4 (± 0.4) | 79.4 (± 0.3) | 53.5 (± 0.4) | 52.5 (± 0.3) | 52.8 (± 0.2) |
| EDEN | 91.4 (± 0.2) | 89.8 (± 0.2) | 88.3 (± 0.3) | 91.7 (± 0.1) | 89.7 (± 0.1) | 87.8 (± 0.2) | 87.5 (± 0.2) | 82.6 (± 0.4) | 79.8 (± 0.4) | 53.9 (± 0.3) | 53.1 (± 0.2) | 52.9 (± 0.1) |
| FedMRN | 91.8 (± 0.1) | 90.2 (± 0.1) | 88.6 (± 0.2) | 92.2 (± 0.1) | 90.0 (± 0.3) | 88.5 (± 0.2) | **88.3 (± 0.2)** | **84.6 (± 0.2)** | **81.4 (± 0.4)** | 55.4 (± 0.5) | 54.1 (± 0.2) | 53.9 (± 0.4) |
| FedMRNS | **92.0 (± 0.1)** | **90.5 (± 0.2)** | **88.9 (± 0.3)** | **92.4 (± 0.2)** | **90.5 (± 0.3)** | 88.7 (± 0.4) | 88.0 (± 0.1) | 84.5 (± 0.3) | 81.0 (± 0.5) | **56.1 (± 0.5)** | 54.7 (± 0.5) | 54.2 (± 0.3) |

Table 2: Accuracy loss compared to FedAvg.

| | FMNIST | SVHN | CIFAR-10 | CIFAR-100 |
|---|---|---|---|---|
| FedPM | -36.4 | -104.4 | -112.3 | -112.4 |
| FedSparsify | -26.0 | -40.1 | -51.8 | -73.8 |
| SignSGD | -4.7 | -6.8 | -16.5 | -34.1 |
| Top-$k$ | -5.9 | -7.3 | -16.0 | -24.5 |
| TernGard | -2.3 | -2.5 | -5.2 | -7.5 |
| DRIVE | -1.8 | -2.4 | -4.1 | -6.6 |
| EDEN | -1.8 | -2.3 | -3.2 | -5.5 |
| FedMRN | -0.7 | -0.7 | 1.2 | -1.9 |
| FedMRNS | **0.1** | **0.2** | 0.4 | **-0.3** |

cumulative accuracy loss across three data distributions. For space constraints, we present only the convergence curves under the Non-IID-2 data distribution, depicted in Figure 3.

First, we examine the model compression baselines, specifically FedPM and FedSparsify. As shown in Figure 3, they two demonstrate markedly lower convergence upper bounds, with a significant decrease in accuracy compared to FedAvg or other baseline methods. This verifies our discussion in the related work section that excessive model compression curtails the expressive capacity and learning potential. Additionally, FedSparsify generally outperforms FedPM in accuracy. That is, training and transferring binary masks for frozen weights is less effective than just transferring the top 3% of model parameters. This reveals the shortcomings of employing masked noise as weights in FL. Instead, FedMRN shows that it is a better choice to employ masked noise as model updates.

Second, we shall analyze the remaining gradient compression methods. All these techniques entail lossy compression of model updates after local training. The errors caused by post-training compression will reduce accuracy to varying degrees. Table 2 demonstrates that SignSGD and Top-$k$ exhibit comparable accuracy, and they notably lag behind FedAvg, particularly evident in situations with high data heterogeneity. Compared with SignSGD, TernGrad extends the value range of compressed model updates from $\{-1, 1\}$ to $\{-1, 0, 1\}$, slightly improving the accuracy at the expense of higher communication costs. DRIVE and EDEN minimize the compression errors with additional calculations after local training. They slightly improve the accuracy but introduce additional computational latency. We will elaborate on this delay in Section 5.6. Notably, FedMRN outperforms all baselines and is comparable to FedAvg in both accuracy and convergence speed, regardless of data distribution. From the perspective of gradient compression, FedMRN is to map model updates into masked random noise, and this process is indeed lossy. However, a fundamental distinction from above methods is that FedMRN learns to compress model updates during local training. This characteristic guarantees that FedMRN can attain reduced compression errors without necessitating additional computational or communication overhead.

## 5.3 Ablation on Progressive Stochastic Masking

To evaluate the efficacy of *PSM*, we test the accuracy of FedMRN under the Non-IID-2 data distributions without *SM*, *PM*, and *PSM*, respectively. As shown in Figure 4, both *SM* and *PM* are essential

for achieving the remarkable accuracy of FedMRN. Upon closer examination, it becomes apparent that *SM* exerts a slightly more pronounced influence on accuracy.

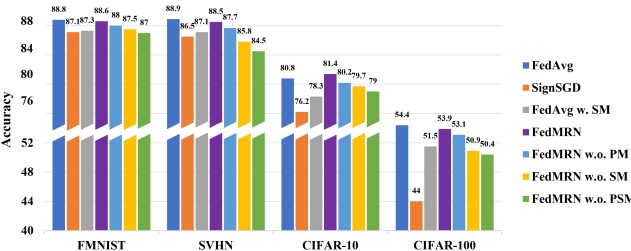

Figure 4: Results of ablation studies.

### 5.4 Comparison with Post-Training Masking

To thoroughly analyze the superior performance of FedMRN, we conduct a comparison with post-training masking. Specifically, we apply stochastic masking on the model updates generated by FedAvg after local training. The sole distinction between [FedMRN w.o. *PM*] and [FedAvg w. *SM*] lies in the timing of masking: during or after local training. As illustrated in Figure 4, the accuracy of [FedMRN w.o. *PM*] notably surpasses that of [FedAvg w. *SM*]. This highlights the significant advantage of incorporating masking during local training as opposed to post-training masking. Furthermore, FedMRN consistently outperforms SignSGD, even in the absence of *PSM*. This further underscores the superiority of learning to compress model updates as opposed to post-training compression.

### 5.5 Impact of the Random Noise

By default, the random noise is uniformly distributed within the intervals [-1e-2, 1e-2] and [-5e-3, 5e-3] for FedMRN and FedMRNS, respectively. Here, we examine the impact of the noise distribution and magnitude using CIFAR-10 under the Non-IID-2 data distribution. Specifically, we investigate the following distributions: Uniform $[-\alpha, \alpha]$, Gaussian $\mathcal{N}(0, \alpha)$, and Bernoulli $\{-\alpha, \alpha\}$. The noise magnitude $\alpha$ is tuned among {6.25e-4, 1.25e-3, 2.5e-3, 5e-3, 1e-2, 2e-2}. As shown in Figure 5, the noise distribution has little impact on performance. The primary factor influencing accuracy is the magnitude of the noise. Besides, our observations indicate that FedMRNS typically requires less noise compared to FedMRN. Specifically, FedMRN achieves comparable accuracy to FedAvg when the noise amplitude falls within {2.5e-3, 5e-3, 1e-2}, and FedMRNS achieves this when the noise amplitude falls within {1.25e-3, 2.5e-3, 5e-3}. The noise required by FedMRN is roughly twice that of FedMRNS. This is intuitively sensible, as $\mathcal{G}(s) \odot \boldsymbol{m}_s = 2\mathcal{G}(s) \odot \boldsymbol{m} - \mathcal{G}(s)$, where $\boldsymbol{m}_s \in \{-1, 1\}^d$ and $\boldsymbol{m} \in \{0, 1\}^d$.

### 5.6 Training Complexity.

Here, we discuss the local training complexity for various methods. Specifically, we measured the local training durations of different methods and their time taken to acquire compressed model updates. The data plotted in Figure 6 is the average of 10 measurements. As shown in Figure 6, FedMRN, FedPM and FedSparsify change the local model structure and slightly increase the local training

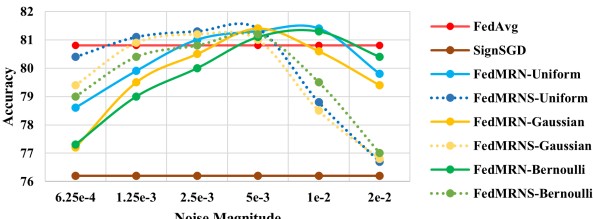

Figure 5: The accuracy of FedMRN with different random noise. The horizontal axis represents the noise magnitude.

time, which is negligible. Regarding the time taken for compressing model updates, both EDEN and DRIVE evidently demand a longer duration, up to one third of local training time. They reduce the compression errors at the expense of extra computational overhead. Differently, FedMRN utilizes the local training process to reduce the compression errors. In summary, FedMRN introduces negligible additional training time to the FL process.

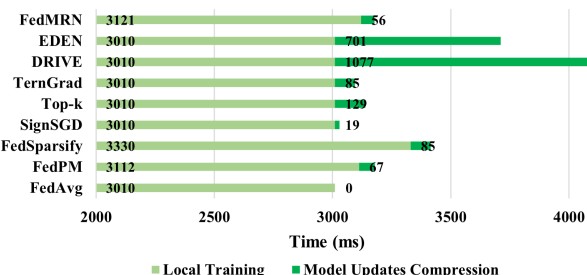

Figure 6: Local training complexity for various methods.

## 6 CONCLUSION AND TAKEAWAYS

In this paper, motivated by the existence of supermasks, we propose to find optimal model updates within random noise. To this end, we propose FedMRN, a novel framework for communication-efficient FL. FedMRN enables clients to learn a mask for each model parameter and generate masked random noise to serve as model updates. To find the optimal masks, we further propose an advanced mask training strategy, called progressive stochastic masking. FedMRN has been fully verified both theoretically and experimentally. The results show that FedMRN is significantly better than relevant baselines and can achieve performance comparable to the FedAvg.

Our experiments and analysis suggest that masked random noise can serve as a viable alternative to model updates. It is preferable to employ the masked noise as model updates rather than as model parameters. Upon further thoughts on FedMRN, we discover a flaw in current methods of compressing model updates, that is the post-training manner. FedMRN attempts to compress model updates into masked noise during local training. It has proved that learning to compress model updates yields superior results compared to post-training compression. We believe this concept has broad applicability in FL and deserves further exploration.

## ACKNOWLEDGEMENTS

This work is supported in part by National Natural Science Foundation of China under grants 62376103, 62302184, 62206102, Science and Technology Support Program of Hubei Province under grant 2022BAA046, and Ant Group through CCF-Ant Research Fund.

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
