# OpenReview forum: "Masked Random Noise for Communication-Efficient Federated Learning"
_acmmm.org/ACMMM/2024/Conference — MM2024 Poster_

### Official Review · Reviewer_ZbUF · 2024-05-24

**Rating:** 4
**Confidence:** 3

**Summary:**

This paper introduces a novel framework called Federated Masked Random Noise (FedMRN) to enhance communication efficiency in federated learning. FedMRN aims to reduce communication overhead by having clients compute optimal model updates within predefined random noise, leading to a significant compression in uplink communication. The proposed framework utilizes 1-bit masks learned by clients to represent model updates as masked random noise and introduces a progressive stochastic masking strategy for mask training. Theoretical convergence guarantees for FedMRN are provided under both strongly convex and non-convex assumptions.

**Strengths:**

- This paper is easy to follow with clear motivations and well-organized structure.
- Experimental results on four datasets show that FedMRN outperforms relevant baselines in terms of convergence speed and test accuracy while achieving similar accuracy as FedAvg
- Sufficient convergence analysis to give theoretical guarantees for the proposed method.

**Limitations:**

- There are some typos in this paper. e.g., line 135 model updates rather than mode.
- This paper needs some explicit quantity analysis of the communication overhead between FedMRN and other baselines since this is the main contribution of this paper.
- How about the cross-silo experiments like ten clients with a 50% sampling rate and other severe non-iid scenarios like the Dirichlet parameter to 0.1?
- Why not compare the performance with FedSGD, FedSGD uploads the gradient of the model and then aggregates and updates the global model, which is more similar to the way of obtaining the global model in this paper. As far as I know, FedAvg directly aggregates the model parameters instead of updating the global model with the uploaded information.

**Suitability:**

3

---

### Official Review · Reviewer_LcgW · 2024-05-25

**Rating:** 4
**Confidence:** 4

**Summary:**

The paper introduces "FedMRN," a novel framework for Communication-Efficient Federated Learning (FL), which addresses the challenge of high communication overheads in FL while preserving data privacy. FL enables collaborative training of machine learning models across distributed clients without centralizing data, but it often incurs significant communication costs. FedMRN requires clients to find optimal model updates relative to global model parameters within predefined random noise. It does this by having clients learn a 1-bit mask for each model parameter and apply masked random noise (the Hadamard product of random noise and masks) to represent model updates.

**Strengths:**

The work has the following advantages.


S1. The authors propose a framework including an advanced mask training strategy called progressive stochastic masking (PSM). This strategy involves stochastic and progressive components to determine the dynamic application of masks to model updates during local training. In practice, clients only need to send their local masks and a random seed to the server after local training, reducing communication overhead.

S2. Theoretical convergence guarantees are established for FedMRN under strongly convex and non-convex assumptions, showing a convergence rate comparable to uncompressed methods like FedAvg. Empirical evaluations on four datasets demonstrate that FedMRN outperforms relevant baselines regarding convergence speed and test accuracy while achieving accuracy levels similar to FedAvg.

S3. The key distinction of FedMRN from existing methods is in using masked random noise as representations of model updates during local training rather than as final model parameters. It also critiques post-training compression strategies and suggests that learning to compress during local training yields better outcomes. This novel concept has implications for improving communication efficiency in FL and invites further research in this direction.

**Limitations:**

However, although the work proposes a masking method to alleviate the intense communication load in FL training, it shows the following areas for improvement. Appropriately addressing the following questions would increase the score.

W1. The author mentions super-masking in related work but needs to include an explanation of super-masking in the experimental and theoretical parts. What is the underlying reason contributing to the super-masking in the proposed methods?

W2.  Is there any difference in convergence analysis between the two masking methods, including Stochastic Masking (SM) and Progressive Masking (PM)? The theoretical results in Sect.4. seem to be standard. Could the author elaborate more on this?

W3. What is the function of Hadamard Product since it appears both in SM and in PM?

**Suitability:**

2

---

### Official Review · Reviewer_iaZ8 · 2024-05-26

**Rating:** 4
**Confidence:** 2

**Summary:**

This paper enhance the communication efficiency in FL by proposing a novel framework that enables clients to learn a 1-bit
mask for each model parameter and apply masked random noise to represent model updates. Progressive stochastic masking strategy is proposed for feasible training and numerical results verify the effectiveness.

**Strengths:**

1. A new framework for communication-efficient distributed training over decentralized data for both IID and non-IID settings, the numerical results demonstrate that it can achieve the similar performance as the training without gradient compression. Ablation study and dicussions are sufficient.

2. The paper is well organized and clearly written, the theoretical and experimental parts seem to be good. The convergence analysis is not so that novel technically but is well adopted to the proposed framework.

**Limitations:**

1. Experiments only conducted on simple image classification tasks, the effectiveness of the proposed framework on other kinds of learning tasks is unclear, as complicated vision tasks may suffer from injected noise. Moreover, classification tasks only require the features extracted by the networks can be seperated for different categories. Could you discuss the potential of applying this training framework to other tasks?

2. Error-compensation techniques [1] are usually applied in distributed training with gradient compression, which can alleviate the performance drop, however the paper did not consider that in the compared baselines. Is there any reason that this error-compensated variants not being considered?

[1] Error Feedback Fixes SignSGD and other Gradient Compression Schemes, ICML 2019.

3. The section of related work is too long as a conference paper.

**Suitability:**

2

---

### Meta-Review · Area_Chair_Zbrh · 2024-06-30

**Recommendation:** Accept (Poster)
**Confidence:** 5

**Metareview:**

The paper introduces a framework for addressing the challenge of high communication overheads in federated learning while preserving data privacy.

In reviews, reviewers acknowledge the clear motivation, originality of the methods, theoretical analysis, and sufficient evaluation and discussion. After the rebuttal, the reviewers acknowledge that all their concerns have been well addressed. Currently, all reviewers vote for positive. Therefore, the AC suggests accepting this paper. The authors should carefully integrate their responses into the camera-ready version.